# A Novel Method for Quantifying Plant Morphological Characteristics Using Normal Vectors and Local Curvature Data via 3D Modelling—A Case Study in Leaf Lettuce

**DOI:** 10.3390/s23156825

**Published:** 2023-07-31

**Authors:** Kaede C. Wada, Atsushi Hayashi, Unseok Lee, Takanari Tanabata, Sachiko Isobe, Hironori Itoh, Hideki Maeda, Satoshi Fujisako, Nobuo Kochi

**Affiliations:** 1Breeding Big Data Management and Utilization Group, Division of Smart Breeding Research, Institute of Crop Science, National Agriculture and Food Research Organization (NARO), Tsukuba 305-0856, Japan; wadak344@affrc.go.jp (K.C.W.); hiroitoh@affrc.go.jp (H.I.); 2Research Center for Agricultural Robotics, Core Technology Research Headquarters, NARO, Tsukuba 305-0856, Japan; hayashia566@affrc.go.jp (A.H.); unseok.lee@naro.affrc.go.jp (U.L.); 3Department of Frontier Research Plant Genomics and Genetics, Kazusa DNA Research Institute, Kisarazu 292-0818, Japan; tanabata@kazusa.or.jp (T.T.); sisobe@kazusa.or.jp (S.I.); 4Center for Seeds and Seedlings, Nishinihon Station (NARO), Kasaoka 714-0054, Japan; hmaeda@affrc.go.jp (H.M.); fujisakos962@naro.affrc.go.jp (S.F.); 5R&D Initiative, Chuo University, Kasuga, Tokyo 112-8551, Japan

**Keywords:** 3D, point cloud, measurement, visual assessment, normal vector, local curvature, compactness, phenotyping, leaf lettuce (*Lactuca sativa*), blistering

## Abstract

Three-dimensional measurement is a high-throughput method that can record a large amount of information. Three-dimensional modelling of plants has the possibility to not only automate dimensional measurement, but to also enable visual assessment to be quantified, eliminating ambiguity in human judgment. In this study, we have developed new methods that could be used for the morphological analysis of plants from the information contained in 3D data. Specifically, we investigated characteristics that can be measured by scale (dimension) and/or visual assessment by humans. The latter is particularly novel in this paper. The characteristics that can be measured on a scale-related dimension were tested based on the bounding box, convex hull, column solid, and voxel. Furthermore, for characteristics that can be evaluated by visual assessment, we propose a new method using normal vectors and local curvature (LC) data. For these examinations, we used our highly accurate all-around 3D plant modelling system. The coefficient of determination between manual measurements and the scale-related methods were all above 0.9. Furthermore, the differences in LC calculated from the normal vector data allowed us to visualise and quantify the concavity and convexity of leaves. This technique revealed that there were differences in the time point at which leaf blistering began to develop among the varieties. The precise 3D model made it possible to perform quantitative measurements of lettuce size and morphological characteristics. In addition, the newly proposed LC-based analysis method made it possible to quantify the characteristics that rely on visual assessment. This research paper was able to demonstrate the following possibilities as outcomes: (1) the automation of conventional manual measurements, and (2) the elimination of variability caused by human subjectivity, thereby rendering evaluations by skilled experts unnecessary.

## 1. Introduction

Physical contact that occurs during the manual measurement of plants can suppress plant growth [1]. Therefore, it is desirable to use noncontact and nondestructive methods to accurately measure plant growth. Noncontact and nondestructive image-based analyses of plant morphology are continually advancing. Data obtained from image analysis technology, such as quantitative characteristic loci (QTL) analysis and genome-wide association studies (GWAS), have been used in molecular genetics [2,3] and is beginning to be used for growth monitoring and growth prediction in crop cultivation [4,5,6,7,8]. Furthermore, image-based 3D measurement methods are also being developed and researched [9,10,11,12]. Notably, 3D measurement is high throughput and can capture a large amount of information with a single measurement [13,14]. Traditionally, plant characteristics have been measured and evaluated manually by ruler or by visual assessment. The former is a labour-intensive and time-consuming process, especially when multiple measurements and plants are involved. The latter method also presents issues such as inter-observer variability and a requirement for skilled evaluators. Our research aims to address these challenges by noncontact 3D analysis of a wide variety of plants to reduce the labour required for measurement and to quantify subjective human evaluation. Therefore, we conducted a review of noncontact 3D measurement methods and developed a highly accurate all-around 3D modelling system to record plant growth [11]. This device uses images to perform 3D modelling and can be applied to various sizes of plants depending on the camera configuration. In this study, we developed a compact device that can be used even in a plant growth chamber that can measure plants ranging in size from a few millimetres to approximately 40 cm. To verify the possibility of measuring plants through 3D modelling, we first chose lettuce, which is easy to obtain and relatively easy to cultivate. Previous studies on the 3D analysis of lettuce have included 3D reconstruction using Kinect [15], analysis of light use efficiency using ray tracing with a handheld laser [16], growth prediction models using 2D projections obtained from 3D models [17], and estimation of harvest yield using volume estimation with 3D models [18]. However, as of now, a 3D measurement method for morphological features such as blistering or undulation (wavy morphology) of leaves has not been established.

There are several types of lettuce, including leaf and head types. Leaf lettuce was selected for analysis in this study. Leaf lettuce is one of the representative vegetables cultivated in plant nurseries. There are many varieties of leaf lettuce, and they have various characteristics, such as large or small wavy leaves or flat leaves, depending on the variety. For lettuce leaves, genes involved in serration have been reported, and these are also associated with susceptibility to tip burn [19]. Furthermore, photomorphogenesis, that is, the phenomenon of plant morphology being controlled by light conditions, has been well studied [20]. In leaf lettuce, it has been reported that light quality and quantity affect leaf shape and colouration [21]. It is important to accurately identify changes in lettuce growth to control for optimal growing conditions.

In Japan, there are 62 evaluation characteristics for lettuce included in the Lettuce Variety Inspection Standards (LVIS; September 2022, Ministry of Agriculture, Forestry and Fisheries) [22] and 55 characteristics in the test guidelines for the International Union for the Protection of New Varieties of Plants (UPOV) [LACTU_SAT *Lactuca sativa* L. TG/13/11 Rev. 2:2017-04-05] [23]. The Japanese National Test Guidelines include some characteristics that overlap with those of the UPOV. Of the 62 LVIS characteristics, 42 are related to appearance, excluding characteristics related to disease resistance and pest resistance. In this study, we mainly examined characteristics adopted in both the Japanese National Test Guidelines and the UPOV Test Guidelines. notably, we focused on morphological characteristic evaluation using 3D analysis and excluded seven characteristics related to colour and five characteristics related to stems. In addition, this study focused on the characteristics that appeared from seeding to harvest in leaf lettuce. As a result, we focused on the 21 characteristics shown in Table 1 that could be analysed using the 3D model.

The characteristics described as “Measure” in the “Survey method” column of Table 1 can be measured with the reconstructed 3D model. Specifically, these are four characteristics encompass the length and volume of each part (Lettuce Variety Inspection Standards: 2, 4, 19, 20; UPOV characteristic 2). The measurement accuracy of our system was evaluated for characteristics related to these items, such as plant size, vertical projected area, and volume measurement.

The characteristics described as “Visual assessment” in the “Survey method” column of Table 1 are qualitative and are evaluated based on subjective human senses. However, there is a possibility that such characteristics can be quantified. We estimated that the characteristics denoted with ** marks in the table could be quantified with 3D models. The characteristics denoted with * marks were by combining AI and recognition technologies.

In addition, seven characteristics related to visual assessment (Lettuce Variety Inspection Standards: 22–28; UPOV: 18–24) were quantified with our newly proposed local curvature (hereinafter referred to as LC) method using normal vectors. Additionally, we examined two characteristics related to blisters on leaves. Then, we confirmed the accuracy of the 3D models by comparing the measured data with the visual assessment results.

In this paper, we describe these results and propose the use of a 3D measurement method for further assessment. Specifically, we develop a new parameter to quantify the blistering of leaves using normal vectors and LC data. Additionally, the growth process of leaf lettuce is quantified using length, projected area from above, volume, and compactness as parameters.

## 2. Materials and Methods

### 2.1. Overview of the 3D Modelling System

Figure 1a shows the 3D modelling system that we previously developed and that was used in this study [24]. The system consists of four cameras, LED lighting, a turntable, measuring posts, a blue background, and a control PC. The system was designed to be used in a compact plant growth chamber (external dimensions of 70 × 70 × 75 cm). The cameras used were Lucid Triton TRI089S-CC (camera body 29 × 29 × 45 mm, 4096 × 2160 CMOS sensor resolution). A wide-angle 8 mm lens was used to capture the turntable and measurement post within the field of view. The shooting time per sample was approximately 7 min. The background colour of the device was painted blue to eliminate the background. The images taken with this device were processed using a closed-loop coarse-to-fine method (CLCFM) that we developed by improving the structure-from-motion/multiview stereo (SfM/MVS) method [25]. Camera positions were estimated by CLCFM, and background and noise were removed by the multimask matching method to obtain high-density 3D point cloud data (Figure 1b).

### 2.2. Basic System Performance

#### 2.2.1. Shape Measurement Accuracy

We have previously reported that this system can measure the shape of a plant body up to approximately 1 mm accuracy for objects up to approximately 1 m in size [24]. Additionally, we have also reported that compared to the accuracy of a handheld laser scanner, the accuracy of this system in surface measurement of fruits is within 1 mm [26]. At this stage, we conducted an evaluation of the accuracy of the angle estimation for the detected surface by using the normal vectors (angle accuracy of the obtained point cloud) that could be obtained from the 3D point cloud generated in the present study.

#### 2.2.2. Angle Measurement Accuracy

To confirm the accuracy of angle measurement for the normal vectors obtained by this system, we tilted a flat plate at angles of 0°, 15°, 30°, 45°, and 60° and reconstructed the point cloud under the same conditions as those for the lettuce. The angle of the plate was measured with a digital angle metre to serve as the actual value. We divided the point cloud into 1 mm squares, 5 mm squares, and 10 mm squares on the *x*-*y* plane and estimated the angle for each region. The mean values and standard deviations of the angles are shown in Table 2 (units are in degrees). The standard deviation of the local angle (orientation) of the plate was 1.28–3.72° for a 1 × 1 mm area, 0.99–1.78° for a 5 × 5 mm area, and 0.63–1.2° for a 10 × 10 mm area. These results indicate that the accuracy of angle estimation improves as the observation area is expanded. When estimating the orientation of an entire leaf or a local part of a leaf with this system, the accuracy is approximately 1° in the 5 × 5 to 10 × 10 mm area.

### 2.3. Plant Materials and Cultivation Methods

We tested four varieties of leaf lettuce, namely, Red Cos, Fancy Green, Frillice, and Red Leaf, that are commercially available on the market for the measurements using our device. Seeds were sown in polyurethane foam blocks (2.5 cm × 2.5 cm × 2.5 cm) soaked in tap water. Then, the plants were placed in a plant growth chamber under a white LED light source with a light intensity of 150 μmol m^−2^ s^−1^ and a 16 h photoperiod at 22 °C. The seedlings were grown hydroponically in 1/2 Otsuka A solution (electrical conductivity of 1.35–1.6 dS m^−1^, pH 5.5–6.5, OAT Agrio, Tokyo, Japan) in plastic trays. One week after sowing, the seedlings were transplanted to plug trays (3 cm × 3 cm × 4 cm) and cultivated using a hydroponic system.

### 2.4. Measurement and Analysis Methods for Lettuce Morphology

#### 2.4.1. Shape Measurements

##### Length and Volume

We measured the changes in length and volume representing six characteristics related to dimension using this device, which provided the most fundamental information about the growth status of the lettuce. We captured images of three varieties of leaf lettuce, namely, Fancy Green, Frillice, and Red Leaf, every other day from 16 to 54 days after sowing. Plant height and long and short diameters were measured using a ruler to obtain actual measurements. Plant height was measured from the ground to the top of the lettuce, while the major and minor axis lengths were measured as the longest and shortest distances, respectively, when viewed from above. We measured three plants for each variety and then calculated the average value of the measurements. We also measured the volume and weight of 17 individuals of Red Cos lettuce of different sizes. We harvested the lettuce plants and measured their volume by submerging them in a graduated cylinder filled with water and recording the water level change as the actual value. For 3D measurements, we used the 3D model reconstructed from images captured by this device to measure the plant height, major axis length, and minor axis length of the lettuce. We estimated the volume by approximating the 3D model using four methods: bounding box, convex hull, columnar solid, and voxel. The bounding box was the smallest rectangle in which the 3D model of a lettuce plant could be enclosed (Figure 2a). The convex hull was the smallest convex polyhedron containing the 3D model of a lettuce plant (Figure 2b). The columnar solid was the product of the height of the bounding box and the area of the vertical projection as viewed from above (Figure 2c). The voxel volume was the volume calculated by converting the 3D point cloud data into a collection of small cubes (voxels) (Figure 2d). The voxel size was 0.5 mm per side.

##### Structural Estimation (Compactness)

There is an index called compactness that is calculated by using the two aforementioned values of voxel volume and convex hull volume. By using this, it may be possible to estimate plant structure, harvesting time, etc. The compactness of plants has been evaluated in various ways in the past. In 2D, compactness is defined as the total leaf area divided by the convex hull area [27,28,29]. In 3D, McCormick et al. (2016) [30] measured shoot compactness by measuring the surface area of the smallest convex polyhedron that contained the entire shoot. In another study that was on grapes, a method was proposed to evaluate the structure of grape clusters using the compactness of 3D models [31]. In the case of lettuce, compactness corresponds to the size of the space between lettuce leaves and may be a potential method for evaluating the density of the leaves. In this paper, compactness is defined by the following equation.
compactness = voxel volume/volume of convex hull(1)

From this equation, it can be seen that the closer the value is to 1, the more space is filled without gaps.

#### 2.4.2. Normal Vector-Based Analysis

Leaf lettuce has a lot of varieties available on the market. Lettuce leaves have differences in leaf morphology, such as wrinkles and flatness, and these morphologies differentiate each variety. Wrinkles and flatness are referred to as “blistering” and “size of blisters” in the Lettuce Variety Inspection Standards (Table 1). Therefore, we investigated a quantitative method for evaluating leaf morphology using normal vector information obtained from 3D models.

##### Normal Vectors

Normal vectors are vectors perpendicular to a plane and can be obtained from the point cloud reconstructed using this system. By utilising the direction information of these normal vectors, it is possible to estimate the orientation (angle) of plant leaves. The normal vectors are calculated using a set of points, including the target point and its neighbouring points in the point cloud. That is, the normal vector is calculated by fitting a local surface to the selected neighbouring points, with different sizes such as 3 × 3, 5 × 5, and 7 × 7, depending on the local surface size of the points (Figure 3).

The normal vector was obtained by solving the relevant plane equation:z = ax + by + c,(2)
y = ax + bz + c,(3)
x = ay + az + c(4)
from the selected point on the three-dimensional coordinate, and each coefficient was obtained using the least-squares method.

##### Local Curvature (LC)

Normal vector data have also been used in plant visualisation by Pound et al. (2014) [32]. They used LC calculated via the normal vector data for 3D reconstruction. We surmised that LC could also be used for plant shape evaluation. It is possible that wrinkles in lettuce leaves could be detected as edges. We applied this method to measure the lettuce plants and calculated LC using the following equation.
(5)$$\mathrm{Local}\;\mathrm{curvature}=\sqrt{s_{nx}^{2}+s_{ny}^{2}+s_{nz}^{2}}$$
where $\left(nx,ny,nz\right)$: Normal vector; $s_{nx}$, $s_{ny}$, $s_{nz}$: standard deviation of  nx,ny,nz.

Using the LC value, it is possible to evaluate “blistering” and “size of blisters” (Table 1). As an example, an arbitrarily created 3D model is shown in Figure 4a–c. Figure 4d–f shows images in which differences in the direction of the plane are visualised by colour coding based on the normal vectors. Figure 4g–i shows a heatmap visualisation of the difference in LC calculated using the normal vector data. The areas where the 3D model has peaks and valleys have large LC values and are shown in yellow to red. The edge part indicated by arrows attached to Figure 4e,f is detected at the positions indicated by arrows in the heatmap of h and i. In the case of a highly convex and concave shape, such as that shown in Figure 4f, many regions with high LC values are detected. In such cases, when the distribution of LC values is shown in a histogram, the distribution is skewed towards larger values of LC on the horizontal axis (Figure 4l). On the other hand, when there are few or small concavities and convexities, such as in Figure 4d,e, the distribution of LC values is skewed towards smaller values (Figure 4j,k). Figure 4k shows high values in the series 0–0.05, which is a flat region with little slope variation, shown in black in the heatmap of Figure 4h.

##### Quantification of LC: Centroid of Histogram

As mentioned above, the distribution of LC values tends to skew towards higher values as the surface becomes more uneven. Therefore, it is possible to evaluate the changes in the blistering of lettuce leaves during growth as a change in the skewness of the distribution of LC values. In this study, we attempted to quantify this change by calculating the centroid of the histogram using the following formula.
(6)$${\mathrm{Centre}\;\mathrm{of}\;\mathrm{Gravity}:LC}_{g}=\frac{\sum_{LC=0}^{1}\left(LC\times frequency\right)}{\sum_{LC=0}^{1}frequency}$$

LC: Local curvature.

## 3. Results

### 3.1. Measurement Accuracy

#### 3.1.1. Accuracy of Lettuce Length Measurement

The 62 lettuce characteristics listed in the Japanese National Test Guidelines (April 2022, Ministry of Agriculture, Forestry and Fisheries) [22] include 4 characteristics related to size: Characteristic 2, cotyledon size; Characteristic 4, plant diameter: Characteristic 19, leaf length and Characteristic 20, leaf width. In this study, the following measurements were made and evaluated for plant size, which can be measured nondestructively. Using the 3D modelling system developed in this study, we continuously captured images of three leaf lettuce varieties, namely, Fancy Green, Frillice, and Red Leaf, every other day from 16 to 54 days after sowing and obtained temporal 3D data of the growth of the leaf lettuce. For each of the three varieties, various parameters were measured using the constructed 3D model data and were compared with manual measurements. We investigated the correlation between the values measured on the 3D model and those measured by hand using a ruler for the plant height, major diameter, and minor diameter of the leaf lettuce for the three varieties. As a result, the coefficient of determination of the measured values for plant height and the major and minor diameters were all 0.99 or higher for all three varieties (Figure 5). The differences between the 3D measurement values and hand measurement values were 0.22 ± 1.82 mm to 0.81 ± 2.15 mm (Table 3). The standard deviation of the measurement error was approximately ±1 mm for plant height, while it ranged from ±1 mm to 3 mm for diameter. One of the reasons for the errors in these measurements may be that the errors occurred when the ruler was pressed against the lettuce leaves and that the measurement positions of the manual and computer measurements did not perfectly match. However, since the correlation between the two measurements was high, it should not be considered to be a problem for practical use.

#### 3.1.2. Accuracy of Lettuce Volume Measurement

We investigated four methods for measuring the shape and volume of lettuce using 3D models. These methods are shown in Figure 2: (a) bounding box, (b) convex hull, (c) columnar solid, and (d) voxel volume. Seventeen Red Cos lettuce plants at different growth stages and of different sizes were used as plant material. The coefficient of determination between the volumes measured by the above four methods and the actual volumes measured by submerging the plants in water in a graduated cylinder were 0.845, 0.846, 0.878, and 0.925, respectively (Figure 6a–d). The accuracy of volume measurement was proportional to the computational cost, and the voxel volume method showed the highest correlation. The other methods also provided high correlation for evaluating the growth and characteristics of lettuce.

Voxel-based volume measurements showed the highest correlation but also had the highest computational cost. In this study, we attempted to calculate voxel sizes of 1 mm or less, but even with a high-performance PC (CPU: Intel Core i9-10900K), it was impossible to perform the calculations. The measurement accuracy of convex hull volume and columnar solid volume was lower than that for voxel volume, but their coefficient of determination with actual measurements were above 0.8 to 0.9, which is considered sufficient for evaluating crop growth and characteristics. Furthermore, it is possible to automatically calculate characteristics such as length and width using these methods.

### 3.2. Normal Vector-Based Analysis (Consideration of Quantification of Visual Assessment)

#### 3.2.1. Visualisation

Visualising the normal vector information that represents the orientation of the point cloud on the 3D model can enable more intuitive visualisation and quantification of the blisters. Therefore, a series of 3D data that recorded the morphological changes in lettuce growth from Section 3.1 were used to visualise the normal vector calculated by the formula shown in Section 2.4.2.

##### Visualisation of Normal Vectors

Figure 7 shows a three-colour visualisation of the normal vectors in the x, y, and z directions for a 3D model of Fancy Green at 30 to 54 days after sowing. Blue indicates when the leaf surface is facing straight up (z axis direction), whereas green or red are displayed when it is facing sideways (x, y axis direction). Visualising the normal data makes it easier to visually read the direction the leaf surface is facing and the complexity of the leaf shape.

##### Visualisation of LC

Figure 8 shows 3D models of a single lettuce leaf cut out and visualised with normal vectors and a heatmap of LC. LC values are displayed in a gradient from red to yellow, green, and blue, with the highest values shown in red. Regions with strong blistering are shown in red, while smooth regions are shown in blue.

Next, we visualised the whole lettuce plant in the same way. Figure 9, Figure 10 and Figure 11 show the results of the normal vector analysis for three lettuce varieties (Fancy Green, Red Leaf, and Frillice). (a) 3D model viewed from the side, (b) 3D model viewed from the top, (c) heatmap visualisation of the LC of the 3D model viewed from the top, and (d) image of the normal vector viewed from the top with three colours for the x, y, and z axis directions. In the heatmap of (c) in Figure 9, Figure 10 and Figure 11, the more uneven areas are shown in red to yellow, and the smoother areas are shown in green to blue. Comparing the three varieties, Fancy Green had many flat areas in blue and green from 26 to 44 days after sowing (Figure 9c), while Frillice and Red Leaf had more wrinkled areas in red from 30 days after sowing, and the wrinkles spread to the entire leaf thereafter (Figure 10c). The wrinkles spread to the entire leaf (Figure 10c and Figure 11c).

#### 3.2.2. Quantification of Convexity and Concavity

As mentioned above, differences in the distribution of LC values among the varieties can be quantitatively compared by determining the centroid position of the LC histogram, as shown in Section 2.4.2. The LC data for the three lettuce varieties shown in Figure 9, Figure 10 and Figure 11 were represented in histograms, and their centroids were calculated (Figure 12). When the LC values are skewed towards smaller values, there are many flat areas with small changes in the normal vectors of neighbouring points. In this case, the centroid value is small. On the other hand, when the LC values are skewed towards larger values, there are many areas with large changes in normal vectors, indicating many areas with high convexity and concavity. In this case, the centroid value is large.

As shown in the heatmap of Figure 9c, Fancy Green had many flat areas that were represented in blue and green colours from 26 to 44 days after sowing. In the histogram of LC shown in Figure 12a, the centre of gravity remained at approximately 0.46 to 0.47 from 26 to 44 days after sowing (Figure 12a). Approximately 54 days after sowing, the area with wrinkles, as shown in Figure 9c, expanded, and the centroid of the LC histogram also increased to 0.56. In contrast, Frillice and Red Leaf showed a tendency of an increase in areas with high convexity and concavity, represented in red in Figure 10c and Figure 11c, from approximately 30 days after sowing. At this time, the centroid of the LC histogram increased along with growth, ranging from 0.52 to 0.64 for Frillice (Figure 12b) and from 0.51 to 0.62 for Red Leaf (Figure 12c). These results indicate that changes in the distribution of visualised LCs are consistent with changes in the value of the centre of gravity of the histogram of LCs. This suggests that quantitative evaluation of convexity and concavity by LC is possible.

### 3.3. Time-Series Measurement Evaluation

We examined the quantification of changes in leaf morphology. We then checked whether it is possible to track these changes. Morphological changes during growth were quantified using time series data from Section 3.1, in which lettuce growth was continuously recorded every other day. Using these 3D models, plant height, vertical projected area viewed from above, voxel volume, compactness, and local curvature were measured and compared among the three varieties (Figure 13a–e).

Red Leaf showed a lower trend in plant height change in the early growth stage at approximately 20 days after sowing (Figure 13a). This is because Red Leaf leaves grow horizontally in the early growth stage; thus, the height of the plant increases slowly. As the plants grew, new leaves were stacked upon the old leaves, and the plant height increased; almost no difference was observed among the three varieties at approximately 35 d after sowing (Figure 13a).

The most common method of evaluating lettuce yield is weighing. However, it is desirable to measure lettuce during growth using a nondestructive method. Therefore, the vertical projected area (ratio of leaf area) and volume were measured from the 3D model and evaluated. On the last day of measurement, the lettuce plants were harvested, and the fresh weights were measured. Red Leaf was heavier than Fancy Green, weighing 31.7 ± 0.5 g and 28.3 ± 1.3 g, respectively. As shown in Figure 13c, Red Leaf > Fancy Green was also observed when the volume was evaluated using voxels. However, as shown in Figure 13b, the vertical projected area was Fancy Green > Red Leaf. These results suggest that voxel volume, a three-dimensional evaluation, reflects characteristics more accurately than the vertical projected area, a two-dimensional evaluation. The reason for the opposite relationship between the vertical projected area or voxel volume of Red Leaf and Fancy Green may be related to the morphology of the lettuce leaves. Comparing the leaf morphology of Red Leaf and Fancy Green, the entire leaf of the former was severely crumpled, while that of the latter was less crumpled and had more areas with smooth surfaces. Leaves that are severely crumpled occupy a larger space. This cannot be quantified using vertically projected areas.

Therefore, we evaluated the ratio of leaves occupying the space using compactness (Figure 13d). When compactness was compared, the relationship between the varieties was Red Leaf > Fancy Green (Figure 13d). This indicates that the ratio of leaves occupying space is larger in Red Leaf than in Fancy Green leaves. Furthermore, in order to compare the difference in the crumpling of leaves between the two varieties, the distribution of local curvature values was expressed in terms of the centroid of the LC histogram as described above, indicating that Red Leaf had a larger value than Fancy Green (Figure 13e). This indicates that Red Leaf leaves were more severely crumpled than Fancy Green leaves.

These results suggest that leaves of Red Leaf occupy a higher ratio of space than those of Fancy Green due to their severe crumpling, resulting in a larger volume in relation to the vertical projected area.

Among the three varieties, Frillice appeared smaller in size, which was reflected in the vertical projected area and voxel volume values viewed from directly above (Figure 13b,c). Frillice showed similar changes to Fancy Green, with lower compactness values in the early stages of growth. Subsequently, the value of compactness increased from 35 d after sowing and showed a pattern similar to that of red leaf from 40 d after sowing (Figure 13d). These changes in values may quantitatively characterise the morphological change of Frillice, in which the leaves that develop in the early stages of growth are not crumpled, but severely crumpled leaves begin to appear in the later stages of growth.

The local curvature of Fancy Green tended to be smaller than that of Red Leaf until the plants matured (Figure 13e). This is consistent with the subjective impression that Fancy Green has fewer crumpled areas and flatter areas on its leaves than Red Leaf.

These results showed that the 3D measurement method provides a great deal of information that cannot be obtained from 2D image measurement and enables analysis by quantification of growth characteristics based on time-series information.

## 4. Discussion

As mentioned above, the LC-based characteristic evaluation was shown to be consistent with subjective evaluation. As shown in the Lettuce Variety Inspection Standards (Table 1), characteristics related to leaf concavity and convexity are evaluated by visual assessment. We believe that the LC-based characteristic evaluation described in this paper may be applicable to this. The following preliminary data have been obtained.

### 4.1. Leaf Lettuce Characteristic Evaluation

We evaluated two lettuce characteristics, Characteristic 22, “Leaf blistering”, and Characteristic 23, “Size of blisters”, which are used in the examination for registration of lettuce varieties. We compared the evaluations by visual assessment, which is currently conducted in the Center for Seeds and Seedlings, NARO, with those by 3D measurements. The visual assessment is to be performed in five or three stages according to the Test Guidelines for Lettuce (April 2022; Ministry of Agriculture, Forestry and Fisheries) [22], as shown in Table 4. We checked whether this visual assessment could be quantified by LC.

First, we performed 3D measurements for a single leaf of 7 lettuce types, Lettuce A to G, sold on the market and created 3D models (Figure 14). We extracted the area excluding the marginal part and part of the midrib (the part where the vascular bundles running through the centre of the leaf are concentrated). We performed LC analysis on this extracted area by the dotted line in the 3D model in the upper part of Figure 14. The area was manually cut out. The heatmap in the middle row of Figure 14 shows that the region marked by orange to red corresponds to the area evaluated by human observation as wrinkles. Samples judged as 7 (strong) for blistering by visual assessment showed many areas marked in orange or red, samples judged as 3 (weak) showed few of those areas, and samples judged as 5 (medium) showed a tendency that was intermediate between the two. When we calculated the centroid of the LC histogram, samples judged as 3 (weak) had values of 0.35 and 0.44, samples judged as 5 (medium) had values of 0.46, 0.41, and 0.39, and samples judged as 7 (strong) had values of 0.48 for both samples (Figure 14). For six samples, excluding lettuce B, there was a suggested correspondence between the value of the centroid of the LC histogram and the strength of blistering evaluated by visual assessment. For the exception of lettuce B, the region extraction was considered incomplete. If the periphery and midrib are not excluded, wrinkles other than blistering may be detected incorrectly. In the future, we will accumulate more data and consider the best method for selecting the area.

Using the same LC analysis data, we also attempted to analyse Characteristic 23, size of blisters. When viewed by the human eye, the larger the flat part of the blister, the larger the blister is judged to be. In flat areas, the value of LC is considered small. Therefore, we analysed the percentage of leaves occupied by areas exhibiting each LC value and examined the correlation with the visual assessment (Figure 15). As shown in the heatmap in the middle of Figure 14, the regions shown in blue to green corresponded to areas that were evaluated as smooth by the human eye. Therefore, assuming that regions with LC < 0.5 shown in blue to green are smooth regions, we calculated the percentage of those regions that occupy the leaf. As a result, in samples where the size of blistering was judged as 3, considered small by visual assessment, the regions with LC < 0.5 occupied 74.9% and 70.4%, while in samples where it was judged as 5, considered medium, they occupied 96.4%, 91.3%, 90.8%, 89.5%, and 76.3%, respectively.

From these results, we suggest that there is a correlation between the LC-based evaluation and the visual assessment for Characteristic 22, leaf blistering and Characteristic 23, size of blisters.

In the surveyed samples, there was some correlation between the evaluation based on LC and the observed evaluation, but the evaluation of the size of the convex and concave parts cannot be fully assessed by the evaluation of the flat parts. However, the evaluation criterion only evaluates the convexity (blister), while the LC data are used to calculate the amount of convexity and concavity but not the orientation. This problem can likely be addressed by combining the normal vector data indicating the orientation of the curved surface to distinguish between convex and concave parts.

From these results, the new LC-based characteristic evaluation approach proposed in this paper displays excellent practical utility. However, further validation is needed due to the lack of data in this case. The correspondence between characteristic evaluation using LC data and characteristic evaluation based on human senses must be further investigated.

Some evaluation characteristics are related to the shape of leaf edges, such as Characteristics 24–28, and these characteristics can be applied in modified LC-based analysis.

For example, Characteristic 24, the degree of undulation of margins, can be evaluated in 3D. Characteristics 25, 26, 27 and 28, which require cutting at the tip of the leaf blade, can be evaluated by projecting the leaves onto a flat surface. We believe that these characteristics can also be quantitatively evaluated from the 3D information acquired by our system. In addition, for the evaluation of these characteristics, it is necessary to extract the margin on the apical part, which in our data (Figure 8c) was detected as an area with a large LC value (red area). It is likely that the evaluation of the undulation and incisions of margins can be analysed by performing region extraction of 3D data; this will be a future challenge.

### 4.2. Three-Dimensional Modelling System

The objectives of our 3D modelling system development are (1) to provide a high-precision 3D modelling device that eliminates noise as much as possible without occlusion, and (2) to develop and provide a technology for evaluating plant traits. Therefore, as part of objective (1), we have developed an all-around 3D modelling device based on photogrammetry and computer vision principles. We have previously reported that this device is a highly accurate and robust system, capable of handling occlusions and noise [11,24,25].

The system developed in this study has a measurement accuracy of 1 mm and an angular accuracy of approximately 1°, which is considered sufficiently high to evaluate the morphology of crops such as lettuce. The 3D model created using a previous 3D imaging system contains nonnegligible noise and occlusion [24] and is considered insufficiently accurate, especially for analysis using normal data. However, the accuracy of the 3D model reconstructed using the system developed in this study was sufficiently high; thus, we attempted to quantitatively evaluate the morphology of the plant leaf using normal data. As a result, 3D measurements of leaf lettuce with high accuracy were possible. The parameters that can be measured are the length, vertical projected area, volume, compactness, and the visualisation and quantification of leaf curvature using a newly introduced local curvature based on normal vectors. The evaluation of characteristics such as concavity and convexity of leaf shape (blistering and size of blisters) in leaf lettuce relies heavily on subjective human judgment. Therefore, we propose a novel approach utilising normal vectors and local curvature (LC) data. With this method, it has become possible to quantitatively assess previously subjective evaluations by humans. These results show that it is possible to evaluate characteristics based on visual assessment using the proposed device and evaluation method.

The concavity and convexity of leaves can also be listed in test guidelines for vegetables such as cabbage, broccoli, and cauliflower; therefore, our technique could be applied to these vegetables as well.

This system has already been installed in 10 research institutes in Japan as a 3D modelling device for plants of various sizes. Target plants include strawberries, rice, soybeans, tomatoes, peppers, and cedar. However, these devices have only been tested and verified to be capable of 3D modelling of each plant. We plan to examine the use of these devices in accordance with the characteristics of each plant and to develop measurement algorithms. Among them, the four items evaluated in this paper (Bounding Box, Convex Hull, Columnar Solid, and Voxel) have the potential to be applied to all kinds of plants. Therefore, prior to the development of these items, we focused on lettuce, which is easy to handle, easy to cultivate in a growth chamber, and can be used to obtain a large amount of data, then conducted a measurement test.

Pongplyapaiboon et al. have installed a 3D modelling device utilising our principle in an inspection line of a plant factory focusing on zoysiagrass and are conducting introductory trials. The zoysiagrass measurement algorithm used in their system differs from ours [33].

The basic configuration of this system consists of a camera, a turntable, and a photographing studio. The part of the system that accounts for the cost is the camera. The smallest system can be configured with one camera for small plants, while the largest one uses eight cameras (for plants up to ~2.4 m).

The advantage of this system is that multiple cameras can be installed according to the shape and size of the plant body. However, for large plants of 2 to 3 m in size, about eight cameras are required, which increases the memory capacity for image acquisition and the time required for transfer and analysis. In other words, a system with eight cameras requires more than twice the memory capacity and processing time of a four-camera system. This also increases the cost of the entire system. 

In addition, for systems installed in places where temperatures rise, such as plant factories, the possibility of thermal runaway increases as the scale of the system grows, and system operation becomes unstable.

Since this system rotates the table to take pictures, the effects of shaking due to wind and vibration cannot be ignored. Therefore, the camera takes pictures while the table is stationary to avoid the effects of shaking as much as possible. However, it took 7 min to capture the image, meaning that reducing the capture time is an issue.

Currently, industrial cameras are used (2000 USD/unit), but the cost can be reduced to one-half to one-third if commercial digital cameras are used. Therefore, depending on the size of the target plant, it is possible to reduce the cost and size of this system. The acquired data capacity is approximately 7 GB for one 3D model in bmp file format for a four-camera system. This can be compressed to 1/3 by converting it to a png file. This means that it is possible to handle time-series data.

There is a demand for a system to evaluate only flowers and leaves for traits (target size; 3 cm~20 cm). Since this system can be configured with a low-cost turntable and one or two cameras, it is considered possible to significantly reduce the cost of the system, make it smaller, and make it mobile.

## 5. Conclusions

In this paper, we proposed an algorithm and its principles for measuring the dimensions and evaluating the visual observation of lettuce using the 3D measurement device that we developed. Ultimately, we demonstrated that various characteristics that cannot be assessed through 2D image analysis can be quantified and evaluated using 3D data.

The dimensional measurements of lettuce were made by taking measurements from the reconstructed 3D model. The algorithm for evaluating visual observation was proposed to calculate the normal vector of the leaf from the reconstructed 3D point cloud and use it to calculate the LC to quantify the visual assessment. As a result, we were able to demonstrate the possibility of automating the conventional manual measurement by reconstructing the 3D model of the plant, and furthermore, by quantifying the visual observation, eliminating the variation in human evaluation and eliminating the need for skilled personnel for evaluation.

This system can be utilised not only for research purposes, such as plant phenotyping and biological analysis, but also for applications in cultivation and farming. It enables growth monitoring to predict the harvesting period of crops and facilitate early detection of growth disorders and plant diseases. In addition, in cultivation in artificial environments such as plant nurseries, crop conditions can be precisely monitored, and the accuracy of cultivation environment control can be improved.

However, these methods need to be validated with more data. Such data and subsequent experiments are in preparation.

Recently, omics analyses such as those related to the transcriptome, proteome, and metabolome have been accelerated, and the data produced have accumulated, but the technology needed to collect phenotype information efficiently is still under development, and it has been difficult to quantify the complexity of morphology. However, the morphological analysis of crops using local curvature and compactness by normal vectors attempted in this study succeeded in visualising and quantifying the complex morphological data of concavity and convexity. Phenotype data as objective variables for explanatory variables, such as omics data, will be more enriched than ever before, which will also contribute to improving the accuracy of genetic analysis.

## Figures and Tables

**Figure 1 sensors-23-06825-f001:**
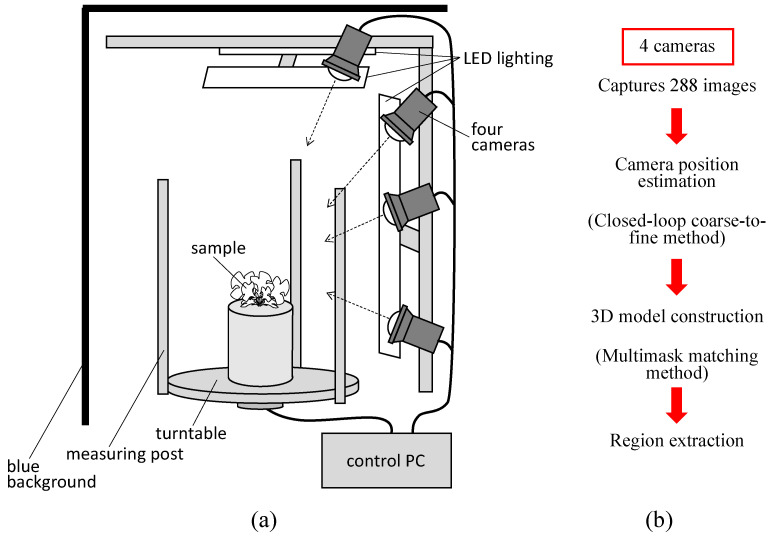
3D modelling devices and measurement schemes. (**a**) All-surrounding-image-capturing device developed in this report. The sample was placed on a turntable and rotated with a motor by 5° to capture images of the sample from all the surrounding directions. (**b**) Overview of 3D modelling and measurement schemes.

**Figure 2 sensors-23-06825-f002:**
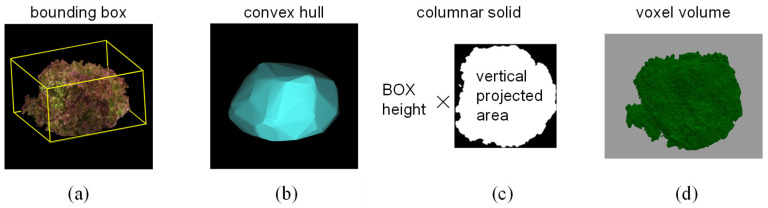
Methods of volume measurement. (**a**): Bounding box, (**b**): convex hull, (**c**): columnar solid, (**d**): voxel.

**Figure 3 sensors-23-06825-f003:**
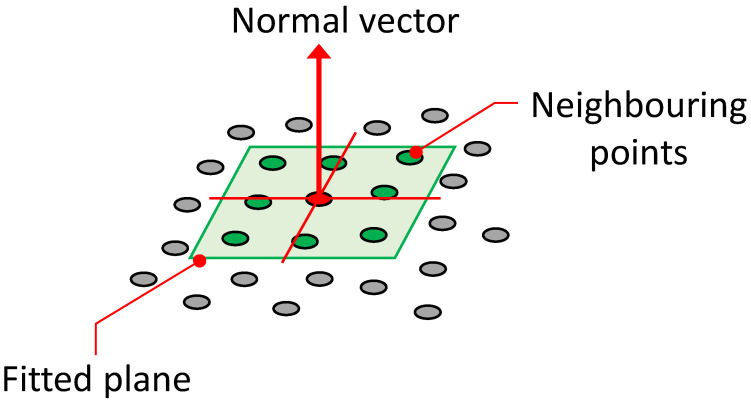
Fitted plane to 3D points and computed normal vector (neighbouring points 3 × 3).

**Figure 4 sensors-23-06825-f004:**
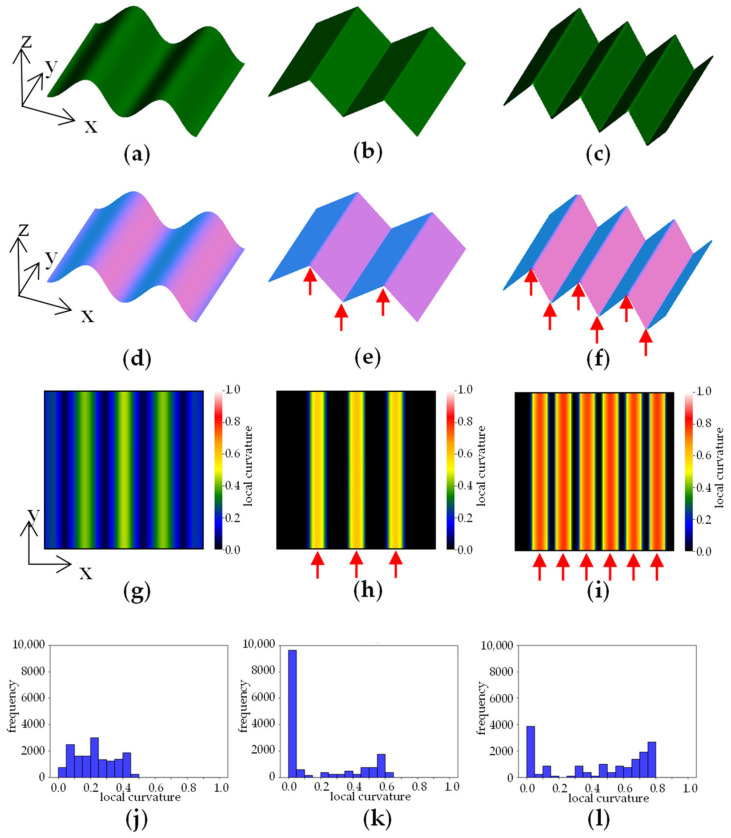
Example of local curvature analysis. (**a**–**c**) 3D model; (**d**–**f**) visualisation of normal vectors; (**g**–**i**) visualizstion of local curvature; (**j**–**l**) histogram of local curvature values. Arrows in the figure indicate edge areas with large local curvature.

**Figure 5 sensors-23-06825-f005:**
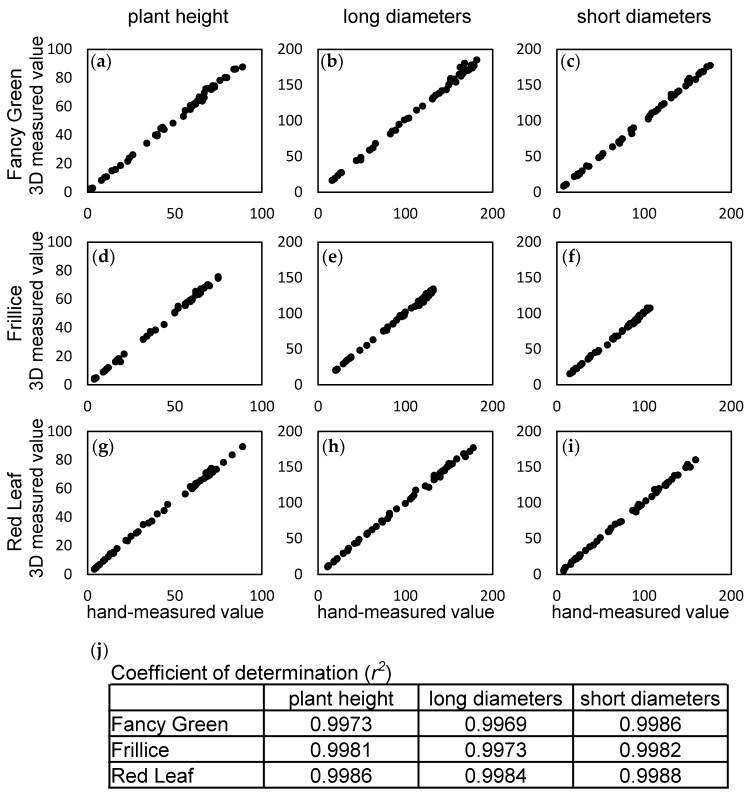
Correlation between 3D measurements and hand measurements. Time-course data for three leaf lettuce varieties, Fancy Green (**a**–**c**), Frillice (**d**–**f**), and Red Leaf (**g**–**i**), were used to determine the coefficient of determination (*r^2^*) between 3D measurements of height, long and short diameter, and hand-measured values (**j**). Values represent the average of *n* = 3.

**Figure 6 sensors-23-06825-f006:**
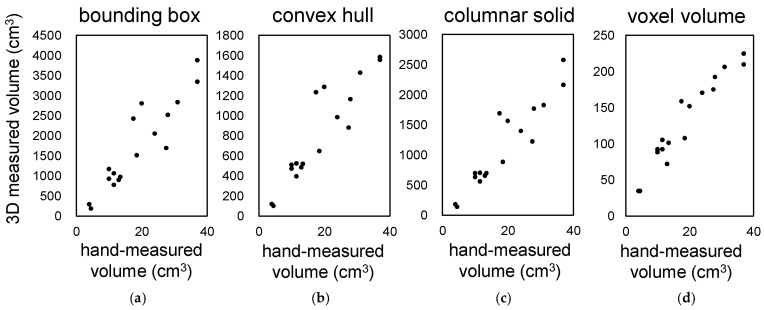
Correlation between hand-measured volume and 3D measurements with our system. Correlation between the volume of the 3D model of lettuce measured by four different methods: bounding box (**a**), convex hull (**b**), columnar solid (**c**), voxel volume (**d**), and hand-measured values using a ruler.

**Figure 7 sensors-23-06825-f007:**
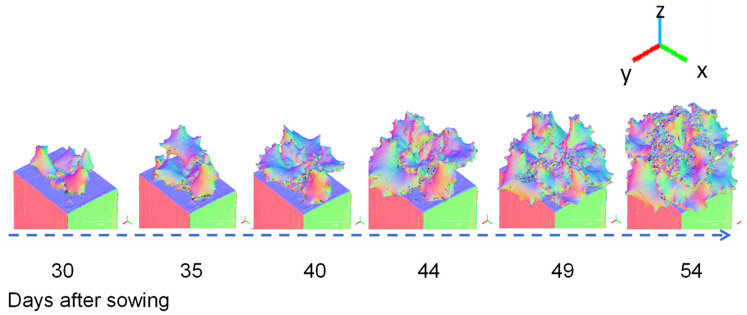
Example of 3D models (Normal Vectors Visualisation). Time-course data of the lettuce variety Fancy Green from 30 to 54 d after sowing were reconstructed using the 3D modelling system developed in this study. Normal vector data in the X, Y, and Z axis directions were visualised on a colour scale.

**Figure 8 sensors-23-06825-f008:**
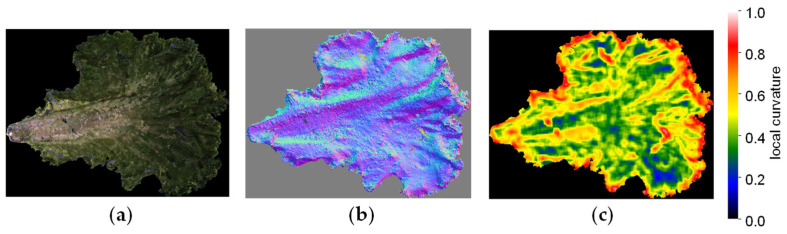
Example of 3D models (local curvature visualisation): (**a**) 3D model of a single lettuce leaf; (**b**) visualisation image of normal vectors; (**c**) visualisation image of local curvature.

**Figure 9 sensors-23-06825-f009:**
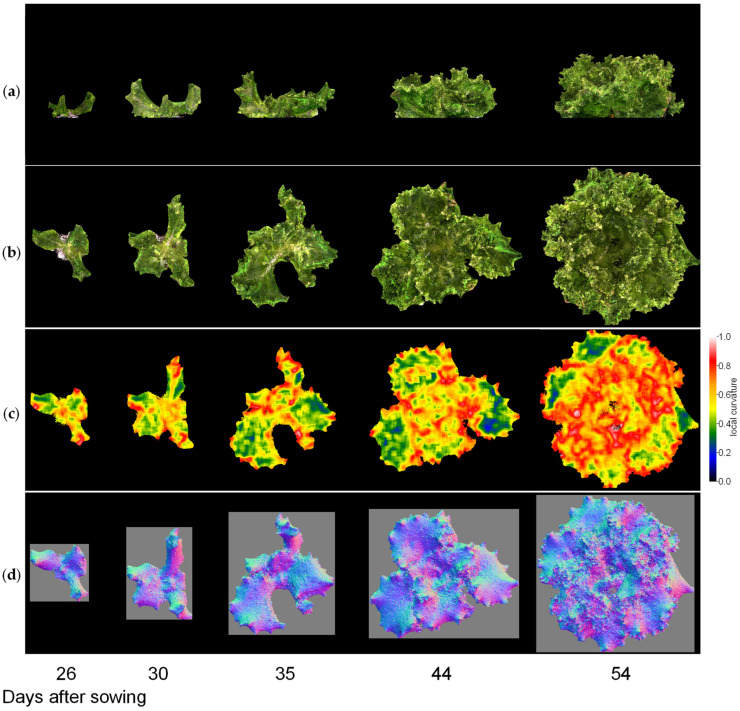
Normal vector analysis of Fancy Green. (**a**) 3D model viewed from the side; (**b**–**d**) 3D model viewed from above; (**c**) local curvature values visualised in a heatmap; (**d**), normal vector visualised in colour scale. The local curvature was calculated using 15 × 15 points.

**Figure 10 sensors-23-06825-f010:**
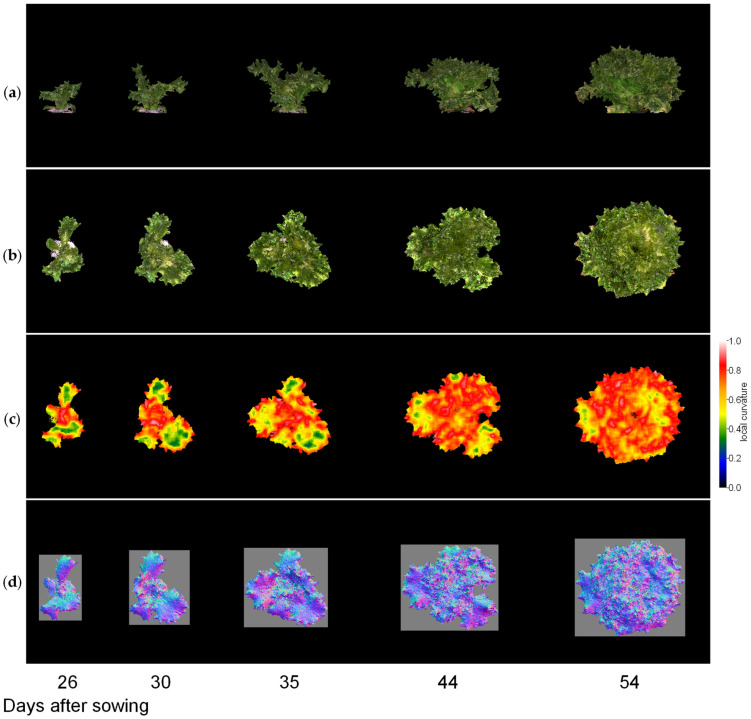
Normal vector analysis of Frillice. (**a**) 3D model viewed from the side; (**b**–**d**) 3D model viewed from above; (**c**) local curvature values visualised in a heatmap; (**d**), normal vector visualised in colour scale. The local curvature was calculated using 15 × 15 points.

**Figure 11 sensors-23-06825-f011:**
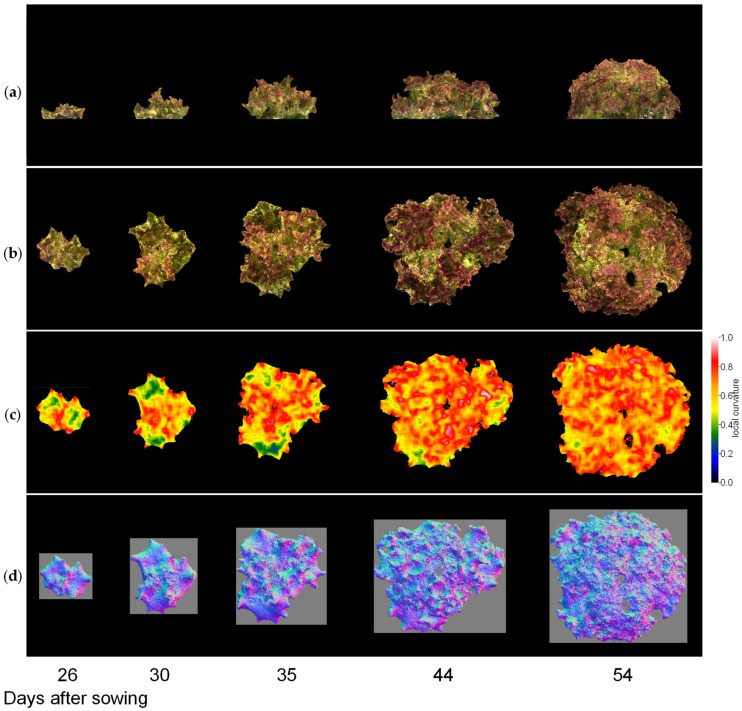
Normal vector analysis of Red Leaf. (**a**) 3D model viewed from the side; (**b**–**d**) 3D model viewed from above; (**c**) local curvature values visualised in a heatmap; (**d**), normal vector visualised in colour scale. The local curvature was calculated using 15 × 15 points.

**Figure 12 sensors-23-06825-f012:**
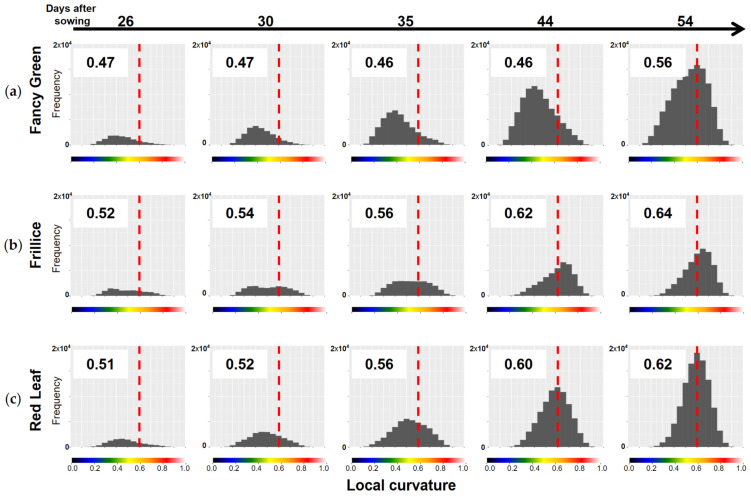
Distribution of local curvature (LC) values for three lettuce varieties. LC was calculated for the top view of the 3D models of three lettuce varieties, Fancy Green (**a**), Frillice (**b**), and Red Leaf (**c**), and the distribution of the values is shown in the histogram. Larger LC values indicate more severe crumpling, whereas smaller values indicate smoother surfaces. The values in the chart represent the centroid of the histograms. The more the centre of gravity of the histogram is skewed to the right, the larger the value is. The colour scale below the horizontal axis corresponds to the colour scale of the heatmap of LC in Figure 9c, Figure 10c and Figure 11c. The dotted line in the figure indicates LC0.6, which is represented by orange on the colour scale.

**Figure 13 sensors-23-06825-f013:**
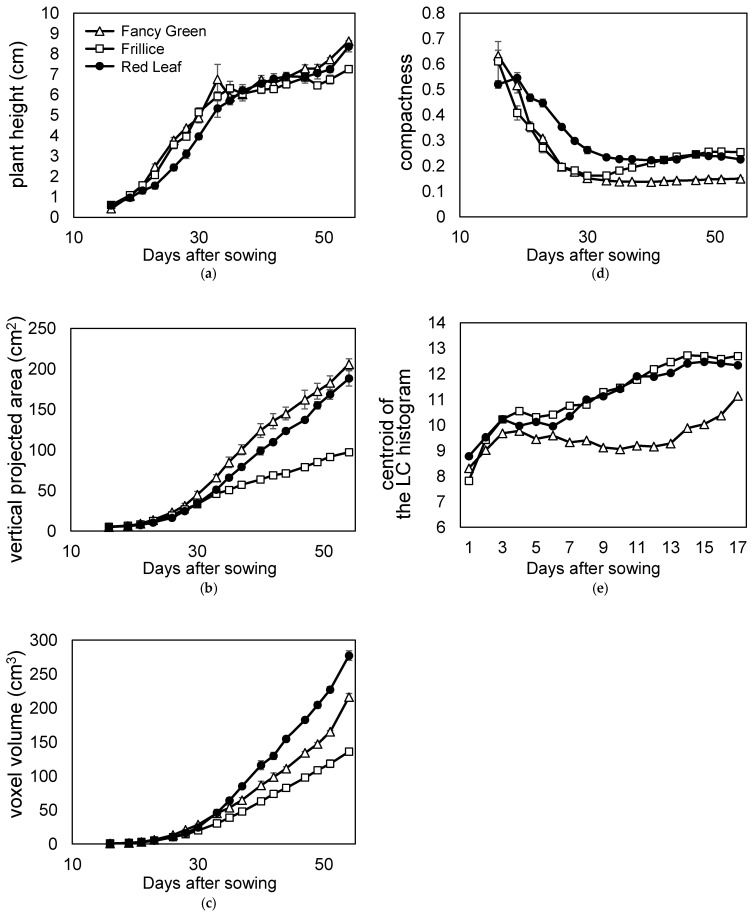
Time course phenotyping of three leaf lettuce varieties. Time course data recorded the growth of three lettuce varieties: Fancy Green, Frillice, and Red Leaf. The 3D model data were reconstructed using the 3D modelling system developed in this study. (**a**) Plant height; (**b**) vertical projected area viewed from directly above; (**c**) voxel volume; (**d**) compactness; (**e**) centroid of the local curvature (LC) histogram.

**Figure 14 sensors-23-06825-f014:**
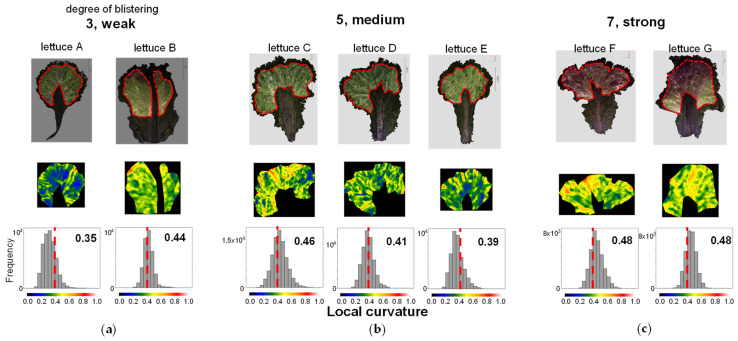
Evaluation of characteristic number 22. Leaf blistering. Top: Colour 3D image of one analysed leaf. The area around the periphery of the leaf and the area excluding the marginal part and part of the midrib are shown enclosed by dotted lines. This region was used to evaluate blistering. Middle: heatmap of the LC of the extracted regions. Bottom: Histogram of LCs in the extracted regions. Leaf blistering of seven lettuce samples (lettuce A to G) were rated “3, weak” (**a**), “5, medium” (**b**), and “7, strong” (**c**) in the prospective evaluation. The numbers in the figure are the values of the centroid of the LC histogram. The colour scale below the histogram also corresponds to each colour in the heatmap. The dotted line in the figure indicates LC 0.4.

**Figure 15 sensors-23-06825-f015:**
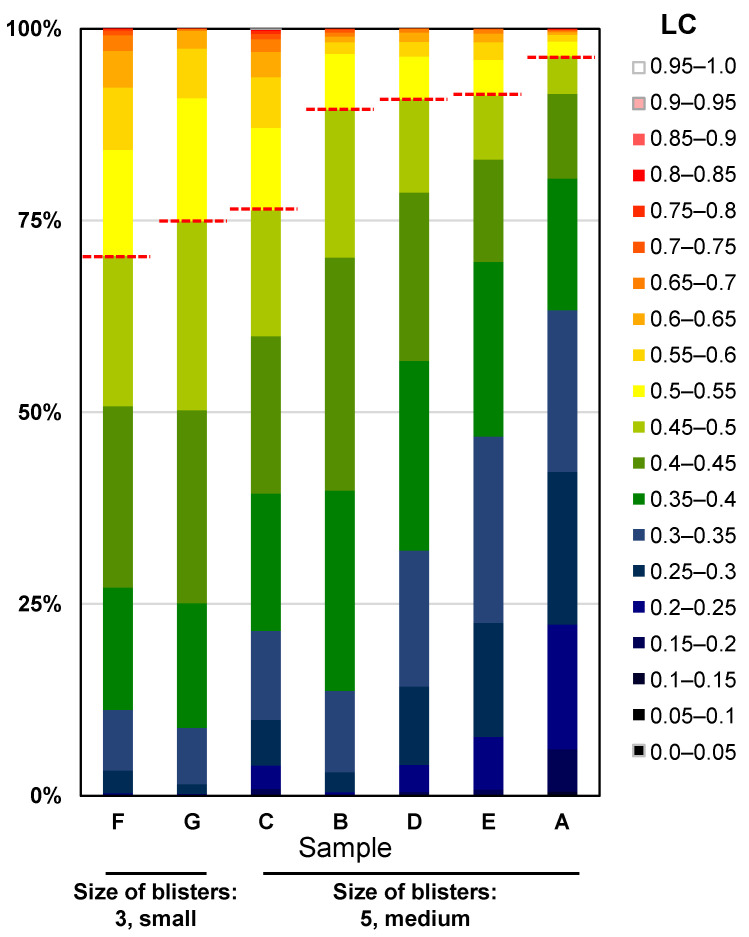
Evaluation of Characteristic 23, size of blisters. Cumulative bar chart of the percentage of area occupied by each LC for leaves rated 3, “small” and leaves rated 5, “medium”, in the prospective evaluation. The colour scale in this figure corresponds to the colour scale of the heatmap in Figure 14. Above the dotted line in the figure indicates a smooth area (LC < 0.5).

**Table 1 sensors-23-06825-t001:** Evaluation characteristics shown in the Lettuce Variety Inspection Standards (April 2012, Ministry of Agriculture, Forestry and Fisheries). ** indicates characteristics that can be measured with the 3D model, and * indicates characteristics that can be determined based on the 3D model and a recognition technique. *** indicates characteristics investigated in this paper.

#	UPOV No.	Characteristic	Survey Method	Possibility of Measurement from 3D Data
	(Accessed on 1 December 2022)			
2		Seedling: size of cotyledon (fully developed)	Measure (mm × mm)	***
3		Seedling: shape of cotyledon	Visual observation	**
4	2	Plant: diameter	Measure (cm)	***
6	4	Leaf: number of leaves	Visual observation/Measure (number)	*
7	5	Leaf: attitude	Visual observation	***
8	6	Leaf: number of divisions	Visual observation	*
9	7	Leaf: shape	Visual observation	*
10	8	Leaf: shape of tip	Visual observation	*
11	9	Leaf: longitudinal section	Visual observation	***
12	10	Leaf: width of lobes	Visual observation	*
19		Leaf: length	Measure (cm)	***
20		Leaf: width	Measure (cm)	***
21	17	Leaf: thickness	Measure (cm)	-
22	18	Leaf: blistering	Visual observation	***
23	19	Leaf: size of blisters	Visual observation	***
24	20	Leaf: undulation of margin	Visual observation	**
25	21	Leaf: type of incisions of margin	Visual observation	*
26	22	Leaf: depth of incisions of margin	Visual observation	**
27	23	Leaf: depth of secondary incisions of margin	Visual observation	**
28	24	Leaf: density of incisions of margin	Visual observation	**
29	25	Leaf blade: venation	Visual observation	*

**Table 2 sensors-23-06825-t002:** Angle estimation. 3D models of flat plates tilted at 0°, 15°, 30°, 45°, and 60° were reconstructed, and angle estimation was performed using their point cloud data. Angle estimation regions were calculated for 1 mm, 5 mm, and 10 mm squares.

		Measured Angle (Average)				
Angle Estimation Area		Angle of Tilted Flat Plate				
		0°	15°	30°	45°	60°
1 × 1 mm	average	−0.34°	14.85°	29.84°	44.52°	59.13°
	SD	1.79	1.39	1.28	2.35	3.72
5 × 5 mm	average	−0.47°	14.78°	29.78°	44.4°	59.09°
	SD	1.78	1.25	0.99	1.35	1.05
10 × 10 mm	average	−0.29°	14.93°	29.88°	44.54°	59.18°
	SD	1.2	0.86	0.76	0.74	0.63

**Table 3 sensors-23-06825-t003:** Difference between 3D-measured and hand-measured values of three lettuce varieties.

	Error of Measurement (mm)		
Variety	Plant Height	Long Diameters	Short Diameters
Fancy Green	0.57 ± 1.41	0.60 ± 3.02	0.81 ± 2.15
Frillice	0.29 ± 1.01	0.22 ± 1.82	0.28 ± 1.23
Red Leaf	0.77 ± 0.92	0.64 ± 2.07	0.72 ± 1.78

**Table 4 sensors-23-06825-t004:** Evaluation criteria for Characteristic 22 (leaf blistering) and Characteristic 23 (size of blisters), cited from the Lettuce Variety Inspection Standards (April 2022; Ministry of Agriculture, Forestry and Fisheries) [22].

#	UPOV No.	Characteristic	Criteria
22	18	Leaf: blistering	1. absent or very weak; 3. weak; 5. medium; 7. strong; 9. very strong.
23	19	Leaf: size of blisters	3. small; 5. medium; 7. large.

## Data Availability

The data presented in this study are available on request from the corresponding author after obtaining permission of an authorised person.

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
