# Peer review of "A Novel Method for Quantifying Plant Morphological Characteristics Using Normal Vectors and Local Curvature Data via 3D Modelling—A Case Study in Leaf Lettuce"

_sensors, 2023, doi:10.3390/s23156825_

Round 1
Reviewer 1 Report
This work constitutes a valuable contribution to the analysis and categorization of plant growth and development parameters, employing a sophisticated image analysis methodology to derive geometric characteristics. The document's organization, proficient use of English, coherent structure, and innovative approach were truly a pleasure to read and review. Furthermore, the presentation of results was executed with excellence style and the images were of good quality and informative.
Based on the above I only have a few comments.
1. How can these image-based data acquisition and support systems be incorporated into traditional production and typing processes? I suggest including aspects in this sense, limitations, advantages, etc.
2. Incorporate some aspects about the cost of potential equipment, capacity for data capture, can it be taken to a commercial system? Is it possible to integrate the system to a mobile device?
3. Try to present figure 13 in another way, pie charts are not widely used.
4. How reproducible could the system be for typing the growth and development of a plant other than lettuce?
5. Under a concept of seeking reproducibility, more transparent and open science, I suggest publishing the codes and images associated with the analysis in a free repository.
I dont have comement about the Ingles style
Author Response
Response to Reviewer 1 Comments
Thank you for your Review.
I attached the revised paper.
Corrections and additions are indicated in red text in revised paper.
Here is how we responded to your comments.
Point1: How can these image-based data acquisition and support systems be incorporated into traditional production and typing processes? I suggest including aspects in this sense, limitations, advantages, etc.
Response1: We are adding a recently submitted paper that describes an actual case study of integration into a production and typing process. References will be added accordingly.
Added text to “5.Conclusion.” as follows; please see 22 Page 618 ~621 line of attached revised paper.
One of these systems is installed in the inspection lines of plant factories. As a system that transports and measures seedlings via mobile robots and conveyors, it is integrated into conventional production and typing processes. This promotes automation from transport to measurement [33].
Added reference [33] as follows; please see 25 Page745 ~746 line of attached revised paper.
[33] Pongplyapaiboon, S.; Tanaka, H.; Hashiguchi, M.; Hayashi, A.; Tanabata, T.; Isobe, S.; Akashi, R. Development of a digital phenotyping system using 3D model reconstruction for zoysiagrass. The Plant Phenome J.2023;6:e20076. doi:10.1002/ppj2.20076.
Point2: Incorporate some aspects about the cost of potential equipment, capacity for data capture, can it be taken to a commercial system? Is it possible to integrate the system to a mobile device?
Response2: Added text to Conclusion. as follows; please see 22 Page 622~23 Page 634 line of attached revised paper.
The basic configuration of this system consists of a camera, a turntable, and a photographing studio. The part of the system that accounts for the cost is the camera. The smallest system can be configured with one camera for small plants, while the largest one uses eight cameras (for plants up to ~2.4 m). Currently, industrial cameras are used (Japanese 300,000en/unit), but the cost can be reduced to one-half to one-third if commercial digital cameras are used. Therefore, depending on the size of the target plant, it is possible to reduce the cost and size of this system. The acquired data capacity is approximately 7 GB for one 3D model in bmp file format for a four-camera system. This can be compressed to 1/3 by converting it to a png file. This means that it is possible to handle time-series data.
There is a demand for a system to evaluate only flowers and leaves for traits (target size; 3cm~20cm). Since this system can be configured with a low-cost turntable and one or two cameras, it is considered possible to significantly reduce the cost of the system, make it smaller, and make it mobile.
Point3: Try to present figure 13 in another way, pie charts are not widely used.
Response3: Figure 13 was changed from a pie chart to a bar chart. Please see Page21 Figure 13. of attached revised paper.
Figure 13. Evaluation of Characteristic 23, size of blisters. Cumulative bar chart of the percentage of area occupied by each LC for leaves rated 3, "small" and leaves rated 5, "medium", in the prospective evaluation. The colour scale in this figure corresponds to the colour scale of the heatmap in Figure 12. Above the dotted line in the figure indicates a smooth area (LC < 0.5).
Point4: How reproducible could the system be for typing the growth and development of a plant other than lettuce?
Response4: Added text to Conclusion as follows;. Please see page 22 615 ~ 618 line of attached revised paper.
This system has already been installed in 10 research institutes in Japan as a 3D modeling device for plants of various sizes. The system has been used to measure not only lettuce, but also strawberries, rice, soybeans, tomatoes, bell peppers, cedar, and many other plants [24].
Point5: Under a concept of seeking reproducibility, more transparent and open science, I suggest publishing the codes and images associated with the analysis in a free repository.
Response5: We are currently preparing a paper summarizing the detailed principles of this device, in which we plan to publish the relevant codes and images.
Thanks for your valuable comments, they are much appreciated.
Reviewer 2 Report
1. Line 177-178, it would be better reduce the subtitle.
2. Pictures should be improved with high resolution.
3. Figure 3 is not meaningful, the basic words should presented in figure.
4. The algorithm and its principles for measuring the dimensions and evaluating the visual observation of lettuce are not clear to presented, please explain with more clear.
5. The results are also not clearly presented, only the measurement results, what is the conclusion after the results?
6. How did the system used to the practice applications since it is too large and high cost?
7. Is this only used for breeding or monitoring in applications.
78. The conclusions and abstracts should be improved further.
Author Response
Response to Reviewer 2 Comments
Thank you for your Review.
I attached the revised paper.
Corrections and additions are indicated in red text.
Here is how we responded to your comments.
Point1: Line 177-178, it would be better reduce the subtitle.
Response1: The subtitle could not be removed due to the structure of the paper,but was changed to bold text instead of sequential numbers to make it easier to read.
Please see attached revised paper;
(before correction)            (after correction)
6 Page 181 line
2.4.1.1. Length and Volume         ➡ Length and Volume.
6 Page 208 line
2.4.1.2  Structural Estimation (Compactness)   ➡ Structural Estimation (Compactness).
7 Page 231 line
2.4.2.1  Normal Vectors  ➡ Normal Vectors.
7 Page 246 line
2.4.2.2  Local Curvature (LC)  ➡ Local Curvature (LC).
8 Page 273 line
2.4.2.3  Quantification of LC: Centroid of Histogram ➡ Quantification of LC: Centroid of Histogram.
Point2: Pictures should be improved with high resolution.
Response2: Replaced all photos (Figure 1.-13.) with higher resolution. Please see Figure1-13 of revised paper.
Point3: Figure 3 is not meaningful, the basic words should presented in figure.
Response3: Basic words are included in Figure 3 for clarity. Please see 7 page Figure3 of attached revised paper.
Point4: The algorithm and its principles for measuring the dimensions/ and evaluating the visual observation of lettuce are not clear to presented, please explain with more clear.
Response4: Added text to Conclusion. as follows; Please see 22 Page 593 ~596 line of attached revised paper.
The dimensional measurements of lettuce were made by taking measurements from the reconstructed 3D model. The algorithm for evaluating visual observation was proposed to calculate the normal vector of the leaf from the reconstructed 3D point cloud and use it to calculate the LC to quantify the visual assessment.
Point5: The results are also not clearly presented, only the measurement results, what is the conclusion after the results?
Response5: Added text to Conclusion as follows; please see 22 Page 596 ~ 600 line of attached revised paper.
As a result, we were able to demonstrate the possibility of automating the conventional manual measurement by reconstructing the 3D model of the plant, and furthermore, by quantifying the visual observation, eliminating the variation in human evaluation and eliminating the need for skilled personnel for evaluation.
Point6: How did the system used to the practice applications since it is too large and high cost?
Response6: Added text to Conclusion as follows; please see 22 Page 615 line ~ 23 page 634 line of attached revised paper.
This system has already been installed in 10 research institutes in Japan as a 3D modeling device for plants of various sizes. The system has been used to measure not only lettuce, but also strawberries, rice, soybeans, tomatoes, bell peppers, cedar, and many other plants [24]. One of these systems is installed in the inspection lines of plant factories. As a system that transports and measures seedlings via mobile robots and conveyors, it is integrated into conventional production and typing processes. This promotes automation from transport to measurement [33].
The basic configuration of this system consists of a camera, a turntable, and a photographing studio. The part of the system that accounts for the cost is the camera. The smallest system can be configured with one camera for small plants, while the largest one uses eight cameras (for plants up to ~2.4 m). Currently, industrial cameras are used (Japanese 300,000en/unit), but the cost can be reduced to one-half to one-third if commercial digital cameras are used. Therefore, depending on the size of the target plant, it is possible to reduce the cost and size of this system. The acquired data capacity is approximately 7 GB for one 3D model in bmp file format for a four-camera system. This can be compressed to 1/3 by converting it to a png file. This means that it is possible to handle time-series data.
There is a demand for a system to evaluate only flowers and leaves for traits (target size; 3cm~20cm). Since this system can be configured with a low-cost turntable and one or two cameras, it is considered possible to significantly reduce the cost of the system, make it smaller, and make it mobile.
Point7: Is this only used for breeding or monitoring in applications.
Response7: Text has been revised and added for clarity as follows; please see 23 page 635 – 638 line of attached revised paper.
This system can be utilized not only for research purposes, such as plant phenotyping and biological analysis, but also for applications in cultivation and farming. It enables growth monitoring to predict the harvesting period of crops and facilitate ealy detection of growth disorders and plant diseases. In addition, in cultivation in artificial environments such as plant nurseries, crop conditions can be precisely monitored, and the accuracy of cultivation environment control can be improved.
Point8: The conclusions and abstracts should be improved further.
Response8-1: The conclusion has been revised and added. Please see 22 page 589 line~23 page 652 line of attached revised paper.
In this paper, we proposed an algorithm and its principles for measuring the dimensions and evaluating the visual observation of lettuce using the 3D measurement device that we developed. Ultimately, we demonstrated that various characteristics that cannot be assessed through 2D image analysis can be quantified and evaluated using 3D data.
The dimensional measurements of lettuce were made by taking measurements from the reconstructed 3D model. The algorithm for evaluating visual observation was proposed to calculate the normal vector of the leaf from the reconstructed 3D point cloud and use it to calculate the LC to quantify the visual assessment. As a result, we were able to demonstrate the possibility of automating the conventional manual measurement by reconstructing the 3D model of the plant, and furthermore, by quantifying the visual observation, eliminating the variation in human evaluation and eliminating the need for skilled personnel for evaluation.
In detail, the system developed in this study has a measurement accuracy of 1 mm and an angular accuracy of approximately 1°, which is considered sufficiently high to evaluate the morphology of crops such as lettuce. The 3D model created using a previous 3D imaging system contains nonnegligible noise and occlusion [24] and is considered insufficiently accurate, especially for analysis using normal data. However, the accuracy of the 3D model reconstructed using the system developed in this study was sufficiently high; thus, we attempted to quantitatively evaluate the morphology of the plant leaf using normal data. As a result, 3D measurements of leaf lettuce with high accuracy were possible. The parameters that can be measured are the length, vertical projected area, volume, compactness, and the visualization and quantification of leaf curvature using a newly introduced local curvature based on normal vectors. Furthermore, we visualized and quantified the curvature of leaves using LC based on the newly proposed normal vectors in this paper. These results show that it is possible to evaluate characteristics based on visual assessment using the proposed device and evaluation method.
This system has already been installed in 10 research institutes in Japan as a 3D modeling device for plants of various sizes. The system has been used to measure not only lettuce, but also strawberries, rice, soybeans, tomatoes, bell peppers, cedar, and many other plants [24]. One of these systems is installed in the inspection lines of plant factories. As a system that transports and measures seedlings via mobile robots and conveyors, it is integrated into conventional production and typing processes. This promotes automation from transport to measurement [33].
The basic configuration of this system consists of a camera, a turntable, and a photographing studio. The part of the system that accounts for the cost is the camera. The smallest system can be configured with one camera for small plants, while the largest one uses eight cameras (for plants up to ~2.4 m). Currently, industrial cameras are used (Japanese 300,000en/unit), but the cost can be reduced to one-half to one-third if commercial digital cameras are used. Therefore, depending on the size of the target plant, it is possible to reduce the cost and size of this system. The acquired data capacity is approximately 7 GB for one 3D model in bmp file format for a four-camera system. This can be compressed to 1/3 by converting it to a png file. This means that it is possible to handle time-series data.
There is a demand for a system to evaluate only flowers and leaves for traits (target size; 3cm~20cm). Since this system can be configured with a low-cost turntable and one or two cameras, it is considered possible to significantly reduce the cost of the system, make it smaller, and make it mobile.
This system can be utilized not only for research purposes, such as plant phenotyping and biological analysis, but also for applications in cultivation and farming. It enables growth monitoring to predict the harvesting period of crops and facilitate ealy detection of growth disorders and plant diseases. In addition, in cultivation in artificial environments such as plant nurseries, crop conditions can be precisely monitored, and the accuracy of cultivation environment control can be improved.
However, these methods need to be validated with more data. Such data and subsequent experiments are in preparation.
Recently, omics analyses such as those related to the transcriptome, proteome, and metabolome have been accelerated, and the data produced have accumulated, but the technology to collect phenotype information efficiently is still under development, and it has been difficult to quantify the complexity of morphology. However, the morphological analysis of crops using local curvature and compactness by normal vectors attempted in this study succeeded in visualizing and quantifying the complex morphological data of concavity and convexity. Phenotype data as objective variables for explanatory variables, such as omics data, will be more enriched than ever before, which will also contribute to improving the accuracy of genetic analysis.
Response8-2:The abstract has been revised and added as follows; please see 1 page 27 ~ 34 line of attached revised paper.
Furthermore, the differences in LC calculated from the normal vector data allowed us to visualize and quantify the concavity and convexity of leaves. This technique revealed that there were differences in the time point at which leaf blistering began to develop among the cultivars. The precise 3D model made it possible to perform quantitative measurements of lettuce size and morphological characteristics. In addition, the newly proposed LC-based analysis method made it possible to quantify the characteristics that rely on visual assessment. This research paper was able to demonstrate the following possibilities as outcomes: (1) the automation of conventional manual measurements, and (2) the elimination of variability caused by human subjectivity, thereby rendering evaluations by skilled experts unnecessary.
Thanks for your valuable comments, they are much appreciated.
Reviewer 3 Report
The article "Morphological Time Course Analysis Using Normal and Local 2
Curvature Information from 3D Modeling Data for Lettuce" is well written
and covers an interesting and useful topic.
The introduction makes a good review of the methods and the problem
under study.
The methodologies are indicated to achieve the proposed objectives.
The results are well presented and discussed.
The conclusions are based on the results and are objective and possibly
useful in more technical and scientific precision agriculture.
Author Response
Point:
The article "Morphological Time Course Analysis Using Normal and Local 2 Curvature Information from 3D Modeling Data for Lettuce" is well written and covers an interesting and useful topic.
The introduction makes a good review of the methods and the problem under study.
The methodologies are indicated to achieve the proposed objectives.
The results are well presented and discussed.
The conclusions are based on the results and are objective and possibly useful in more technical and scientific precision agriculture.
Response:
Thank you for your peer review.
We would like to make this research useful in the field of agriculture. And we would like to develop it further.
A revised version is attached in response to reviewers 1 and 2.
Thanks for your valuable comments, they are much appreciated.

Round 2
Reviewer 2 Report
The paper was revised well.
Author Response
Thank you for your review.